# Febrile-Range Hyperthermia Can Prevent Toxic Effects of Neutrophil Extracellular Traps on Mesenchymal Stem Cells

**DOI:** 10.3390/ijms232416208

**Published:** 2022-12-19

**Authors:** Caren Linnemann, Andreas K. Nussler, Tina Histing, Sabrina Ehnert

**Affiliations:** Siegfried Weller Institute for Trauma Research, BG Unfallklinik Tübingen, Eberhard Karls Universität Tuebingen, 72076 Tuebingen, Germany

**Keywords:** neutrophil extracellular traps, MSCs, differentiation, DNase, febrile-range hyperthermia

## Abstract

Fracture healing is characterized by an inflammatory phase directly after fracture which has a strong impact on the healing outcome. Neutrophils are strong contributors here and can release neutrophil extracellular traps (NETs). NETs are found after trauma, originally thought to capture pathogens. However, they can lead to tissue damage and impede wound healing processes. Their role in fracture healing remains unclear. In this study, the effect of isolated NETs on the function of bone-forming mesenchymal stem cells (SCP-1 cells) was examined. NETs were isolated from stimulated healthy neutrophils and viability, migration, and differentiation of SCP-1 cells were analyzed after the addition of NETs. NETs severely impaired the viability of SCP-1 cells, induced necrosis and already nontoxic concentrations reduced migration significantly. Short-term incubation with NETs had a persistent negative effect on osteogenic differentiation, as measured by AP activity and matrix formation. The addition of DNase or protease inhibitors failed to reverse the negative effect of NETs, whereas a short febrile-range temperature treatment successfully reduced the toxicity and membrane destruction. Thus, the possible modification of the negative effects of NETs in fracture hematomas could be an interesting new target to improve bone healing, particularly in patients with chronic diseases such as diabetes.

## 1. Introduction

Traumatic injuries affect many people within their lifetime. The necessary wound healing is a complex and tightly regulated process. When these interactions do not work properly, chronic wounds can develop, which represent a major clinical problem after tissue injury [1]. Neutrophil extracellular traps (NETs), DNA-protein structures released from neutrophils as a defense mechanism [2], were shown to contribute to this pathological process [3,4]. The released DNA and proteins directly damage tissue and dysregulate the highly orchestrated healing process [5,6]. Especially in diabetic conditions, excessive NET formation was found to severely impair tissue regeneration [7,8] (a detailed review on the role of NETs in wound healing can be found in [9]).

Apart from wounds, trauma is often associated with bone fractures. Bones fractured due to trauma still do not heal properly in up to 10% of all cases [10]. The healing of wounds and fractures follows a similar process [11] and both are characterized by an important inflammatory process at the beginning, where high amounts of neutrophils infiltrate the site of damage [12]. In general, neutrophils form NETs rapidly after exposure to different stimuli, e.g., bacteria or high calcium levels [13]. Their main function is to trap and kill pathogens [14]. Therefore, NETs represent a powerful defense mechanism against invading pathogens in a wound or fracture. Likewise, after a fracture, circulating NETs markers like free DNA and nucleosomes were found to be increased [15,16]. Despite their important role in pathogen defense, NETs could contribute to delayed fracture healing, similar to the detrimental effects seen in wound healing. However, their role in fracture healing has not yet been investigated so far.

The cells mainly responsible for bone healing are mesenchymal stem cells (MSCs). They are attracted by the inflammatory environment of the fracture gap [17]. After migration of MSCs into the fracture gap, cells need to proliferate and differentiate to promote bone healing. In the fracture gap, they come into direct contact with the inflammatory environment and, hence, with released NETs. Thus, NETs could exert strong effects on MSCs. In the context of wound healing, an interaction of MSCs and NETs was already shown [18]. Impaired MSC function, which can be caused by persistent inflammation in the fracture gap [19,20], is associated with delayed bone healing. Similarly, for endothelial cells, where malfunction is associated with nonunion development [21], NETs were already shown to hamper function and proliferation [22]. However, none of these studies analyzed the effect of NETs on MSCs in the fracture gap. Thus, the effect of released NETs on MSCs proliferation and function in the specific context of fracture healing shall be analyzed in this study.

Regarding possible interventions, the several different components of NETs (DNA, proteases, antimicrobial proteins, histones) allow for different target options. In one study, the protein components of NETs were responsible for reduced endothelial functionality [22], whereas in wound healing and bone implant loosening, the application of DNase has been proven to be effective in improving the therapeutic outcome [3,23]. In contrast, in cartilage destruction, neutrophil elastase (NE) was the responsible NETs component and its inhibition rescued cartilage integrity [24]. In general, the prevention of NET formation itself appears to reduce the toxic effects of NETs most effectively [22,25]. However, total NETs depletion is not desirable as NETs represent an important part of the antimicrobial defense [13], and their lack may increase the susceptibility to bacterial infections and sepsis [26].

Considering the current literature and obvious similarities between wound and fracture healing, it is conceivable that excessive NET formation has detrimental effects on bone regeneration. Thus, the aim of this study was to investigate the effects of NETs on mesenchymal stem cells, one of the most important cells for bone regeneration, and to identify possible target mechanisms for problematic fracture healing.

## 2. Results

### 2.1. Isolated NETs Are Toxic to SCP-1 Cells

NETs induced a concentration-dependent effect on cell viability in SCP-1 cells. After 48 h of stimulation, the strongest effects were seen on mitochondrial activity, which was significantly reduced when stimulated with at least 0.125 ng/μL of NETs (Figure 1A). The impaired mitochondrial activity resulted in a dose-dependent increase in LDH release (0.5 ng/μL and 1 ng/μL of NETs, Figure 1C), and finally, a significant reduction of the total protein content at 1 ng/μL of NETs ( Figure 1B). The EC_50_ for NETs was calculated as 0.357 ng/μL for mitochondrial activity, 0.516 ng/μL for total protein content, and 0.555 ng/μL for LDH release. In contrast to NETs, genomic DNA did not affect mitochondrial activity, total protein content, or LDH release.

The results on cell viability were confirmed by an indirect assay for the measurement of membrane integrity (Figure 1D). The addition of NETs significantly induced the release of internalized Calcein into the supernatant when compared to the control cells. Furthermore, the addition of NETs led to high levels of propidium iodide-positive cells (Figure 1F), characteristic for necrotic cell death. Incubation of cells with NETs in PBS (Appendix A) showed a change in morphology already after 90 min. Considering that Calcein release from and propidium iodide uptake in SCP-1 cells occurred quickly (22 h) after incubation with NETs and was faster in PBS, this suggests a physical (e.g., charge-dependent) mechanism of cell death induction.

### 2.2. NETs Impair Migration and Differentiation of SCP-1 Cells

Migration and differentiation of MSCs are crucial for successful fracture healing. Therefore, SCP-1 cells were incubated with different concentrations of NETs, and migration to a cell-free area was analyzed after 45 h. NETs dose-dependently reduced the migration of SCP-1 cells. Migration was reduced by approximately 20% at a concentration as low as 0.0625 ng/μL and by more than 30% at 0.5 ng/μL (Figure 2A). Figure 2B shows the higher amount of cell-free area with increasing NETs concentration, depicting the reduced migration of the SCP-1 cells.

To mimic the inflammatory phase of bone healing, SCP-1 cells were incubated with NETs for 48 h in growth medium. After the removal of NETs, SCP-1 cells were differentiated (Day 0) for 28 days.

As seen before, mitochondrial activity was significantly reduced at the three highest concentrations of NETs (0.125–0.5 ng/μL). This effect persisted until day 3. After day 7, mitochondrial activity has nearly recovered in all conditions (Figure 3A); for the total protein content, the recovery of cells took longer than for the mitochondrial activity. Until day 10, the total protein content was significantly reduced at the two highest concentrations of NETs (0.25 and 0.5 ng/μL). 0.125 ng/μL NETs reduced the total protein content on day 3 significantly (Figure 3B).

AP activity constantly increased with time in differentiating SCP-1 cells. NETs delayed the increase in AP activity, which was significantly reduced with all tested NETs concentrations on days 10 and 14 (Figure 3C), suggesting a prolonged negative effect of NETs on the differentiation of SCP-1 cells. As a result, incubation with 0.5 ng/μL of NETs significantly reduced matrix mineralization, measured on day 28 (Figure 3D). Alizarin Red staining (Figure 3E) showed clear formation of calcium deposits as an indicator of matrix formation in the control conditions. NETs-treated cells, however, showed no signs of calcium deposition.

### 2.3. DNase Is Not Effective in Preventing NETs Toxicity

The most obvious tool to disassemble NETs is DNase treatment [3], thought to destroy the NETs’ backbone. Therefore, the possibility of DNase to reduce or even prevent NET toxicity in SCP-1 cells was examined (Figure 4). The NET-induced decrease in mitochondrial activity could not be attenuated by DNase treatment (Figure 4A), although the DNA content was effectively reduced (Figure 4C). Total protein content revealed that slightly more cells remained attached when SCP-1 cells were additionally treated with DNAse compared to NETs alone (Figure 4B). DNase itself did not affect mitochondrial activity or total protein content of SCP-1 cells.

Enzymes and antimicrobial peptides are the other major effectors of released NETs. The DNA content of the NETs directly correlated with the protein content (Figure 5E). Therefore, SCP-1 cells were incubated together with NETs and proteinase K or different protease inhibitors (Leupeptin/Pepstatin A) to inhibit or destroy those effectors. The addition of proteinase K did not restore mitochondrial activity or total protein content (Figure 5A,B). Similarly, incubation with Leupeptin and Pepstatin A did not prevent toxicity of added NETs in SCP-1 cells (Figure 5C,D). Histones are another major component of NETs. Histone antibody treatment or the addition of heparin as a binding competitor for histones both showed no effect on NETs toxicity (Figure 5F–H).

### 2.4. Temperature Treatment Can Prevent NETs Toxicity

As neither DNase nor protease treatment effectively reduced the toxicity of NETs on SCP-1 cells, an alternative approach was investigated. Since protein denaturation occurs at higher temperatures, we pre-treated NETs with different temperatures and examined the effect on SCP-1 cells (Figure 6).

As before, SCP-1 cells were challenged with native or heat-treated NETs at a concentration of 0.5 ng/μL, known to affect mitochondrial activity, membrane integrity, and total protein content. Pre-incubation of the NETs at 75 °C or 99 °C for 20 or 10 min, respectively, reduced the toxic effect of the NETs (Figure 6A,B). Pre-incubation of NETs at 40 °C (febrile-range) for 60 min was also sufficient to reduce the toxic effects of NETs in SCP-1 cells (Figure 6A,B). LDH release as an indicator for cell death was nearly reversed with 40 °C pre-treatment of NETs (Figure 6C). Calcein leakage was reduced by 40 °C pre-treatment of NETs (Figure 6D). Live–dead staining with Calcein-AM/PI (Figure 6E) showed less dead (necrotic) SCP-1 cells when stimulated with NETs pre-incubated at 40 °C as compared to stimulation with naïve NETs. Interestingly, NETs-treated SCP-1 cells showed increased fluorescence intensity of Calcein staining compared to SCP-1 cells treated with 40 °C NETs, indicating lower stress levels in SCP-1 cells with 40 °C-treated NETs. Pre-incubation of NETs at 40 °C for 30 min also reduced toxicity; however, the effect was less pronounced than with 1 h pre-incubation (Appendix A).

## 3. Discussion

MSCs are one of the most important cell types responsible for tissue regeneration after a fracture. They are normally recruited to the site of damage by cytokines released by immune cells. There is increasing evidence that neutrophils, which are the first infiltrating cells to reach the fracture hematoma, form NETs after trauma [15,16]. However, the effects of NETs on MSC viability and function have not been studied to date. In the present study, we demonstrated potent toxicity of isolated NETs towards bone-forming MSCs.

The role of neutrophils in fracture healing is critically discussed. Invading the site of fracture in large amounts within a few hours after trauma, neutrophils represent the first line of defense against pathogens [27]. They release cytokines essential for the following invasion and activation of immune cells, endothelial cells, and MSCs [12,28]. However, when dysregulated, neutrophils may also exert negative effects on fracture healing [29,30]. In (diabetic) wounds, excessive formation of NETs was related to impaired wound healing. Within wounds, NETs showed direct toxicity towards epithelial and endothelial cells [22]. As NET-markers were shown to be increased also following trauma, toxic effects on bone-resident cells might be conceivable. Indeed, bone marrow-derived SCP-1 cells reacted as being surprisingly sensitive to NETs. In contrast to other studies, only 1/5 of the NETs were required to obtain toxicities comparable to melanoma [31], epithelial, or endothelial cells [22]. SCP-1 cells showed membrane leakiness within hours after NETs exposure, indicating physical damage to the cell walls, resulting in necrotic cell death.

In general, prevention of NET formation would be the most favorable approach, but the positive effects in antimicrobial defense should not be neglected, not to mention that in the case of a spontaneous stimulus of NETs, e.g., trauma, depletion of NETs is impossible. Therefore, controlling the harmful side effects of aberrant NET formation is the only remaining option.

The backbone of the NETs is comprised of negatively charged DNA. Therefore, the most obvious strategy to reduce the toxic effects on bone is the enzymatic digestion of the DNA backbone with DNase [31,32]. In diabetic mice, enzymatic digestion of the DNA backbone of the NETs within the scab improved wound healing [7]. However, in our study, DNase treatment could not abrogate the toxicity of NETs. These results are in line with studies in tissues where DNA was not the toxic component of the NETs [25,33]. DNase treatment even negatively affected healing in a musculoskeletal injury mouse model [6]. The potential negative effect of DNase, however, was not observed in our study.

Considering that the DNA backbone itself is not causing the strong toxicity observed, proteases, antimicrobial enzymes, or histones attached to the DNA backbone might be the toxic component. However, specifically addressing one of these is challenging, as the protein composition of NETs can vary depending on stimuli and the basic constitution of neutrophils [2]. For example, investigating cartilage destruction, NE was identified as the main effector. Inhibition of NE activity with Sivelestat could rescue the destructive effect of NETs on articular cartilage [24]. In this study, the more unspecific protease inhibitors leupeptin and pepstatin A were used. Although, being able to inhibit neutrophil protease activity (e.g., Cathepsins, [34,35]), these two protease inhibitors were ineffective in reducing the toxicity of NETs on SCP-1 cells. This is in agreement with the study that showed that protease inhibition is not sufficient to reduce NET-induced toxicity in endothelial cells [22]. This suggests that either protease inhibition was not efficient or selective enough, or that proteases are not the crucial mediators.

To address the first option, a more general approach was chosen in which the proteins attached to the NETs were digested with proteinase K. This approach was also ineffective in reducing the NET-induced toxicity in SCP-1 cells. The fast induction of membrane leakiness in SCP-1 cells, however, supports the role of one of the many protein components of NETs. Using epithelial and endothelial cells, histones were identified as the most toxic components of the NETs [22]. It has been proposed that the positively charged histones can directly penetrate plasma membranes [36] via interaction with the phospholipids of cell membranes [37]. For instance, Histone H4, one component known to be released with NETs, directly led to pore formation in cellular membranes [38]. By charge interaction, heparin was previously shown to reduce toxicity induced by histones [39,40] but this approach could not effectively reduce toxicity of isolated NETs to SCP-1 cells.

As not one single mediator could be identified, more general approaches were chosen to either degrade or denature the proteins attached to the NETs. Interestingly, protein denaturation with heat could almost totally abrogate the toxic effect of the NETs. The 99 °C treatment was the most effective but cannot be applied to the human body without harming the surrounding tissue. Therefore, lower temperatures were also used. One hour of incubation at 40 °C was sufficient to prevent the toxicity of NETs and reduce the induction of membrane leakiness. Such a short-term incubation at 40 °C is transferable into clinical settings, as it represents febrile-range hyperthermia. In the bone context, for example, the short-term application of such febrile-range hyperthermia has been shown to improve bone formation in a rat fracture model and induce osteogenic differentiation of MC3T3 cells [41]. However, besides possible positive effects on MSCs and tissue-resident cells, the direct effects of febrile-range hyperthermia on neutrophils may not be neglected. In one study, hyperthermia induced higher NETs release and less cytokine production [42], which could be detrimental to MSCs.

In the fracture gap, interaction of MSCs and neutrophils takes place in both directions. Regarding this interaction, MSCs seem to reduce the activation of neutrophils [43] whereas TLR-4-dependent activation of MSCs improves wound healing in a neutrophil/NETs-dependent manner [18]. The single-sided analyses in our study must be seen as a limitation and further research in a more complex model system is necessary to evaluate this effect.

In summary, our data suggest that MSCs are particularly sensitive to NETs, which rapidly induce necrotic cell death in these cells. As a consequence, the migration and differentiation of the MSCs were affected. Therefore, aberrant NET formation in the fracture gap could severely impair the fracture healing process due to impaired MSC function. Febrile-range hyperthermia successfully prevented toxicity and could thus be an option for the development of new fracture treatments in patients.

## 4. Materials and Methods

### 4.1. Human Material

Venous blood was collected from healthy volunteers after informed consent. All experiments were carried out with the permission of the Ethics Committee of the University of Tübingen (Ethics vote number 666/2018B02).

### 4.2. Isolation of Neutrophils and Neutrophil Extracellular Traps

Neutrophils were isolated as previously described [44]. Briefly, blood was separated by density gradient centrifugation using Lympholyte Poly isolation medium (Cedarlane, ON, Canada). After washing twice with PBS (Biochrom, Merck KGaA, Darmstadt, Germany), neutrophils were stimulated for NET isolation as previously described [31]. Neutrophils were seeded at a density of 5 × 10^6^ cells/mL in RPMI plain medium (Sigma-Aldrich, St. Louis, MO, USA) and stimulated with 500 nM PMA (Abcam, Cambridge, UK) for 4 h. NET formation was confirmed by immunofluorescence staining of NETs. Cells were harvested with a cell scraper and collected by centrifugation at 600× *g* for 10 min. The supernatant was transferred to fresh 1.5 mL tubes and centrifuged at 18,000× *g* and 4 °C for 15 min. The formed pellets were resuspended in 10 μL of PBS each. DNA measurement of the pooled samples was performed for quantification of NETs with an LVIS plate in the Omega plate reader (BMG Labtech, Ortenberg, Germany). The protein content was determined by Lowry Assay ([45]).

### 4.3. Culture of and Differentiation of SCP-1 Cells

Human immortalized bone marrow-derived mesenchymal stem cell SCP-1 cells were kindly provided by Prof. Matthias Schieker [46]. Cells were cultured in MEM-α (HiMedia Laboratories, Mumbai, India) with 5% fetal calf serum (FCS, Thermo Scientific, Waltham, MA, USA) and sub-cultured when 80–90% confluent. The medium was changed every 3–4 days. Cells were regularly tested for mycoplasma contamination (MycoAlertTM Mycoplasma Detection Kit, LT07-118, Lonza, Basel, Switzerland). Osteogenic differentiation was induced with MEM-α supplemented with 1% FCS, 200 μM of L-ascorbate-2-phosphate, 5 mM of β-glycerol-phosphate, 25 mM of HEPES, 1.5 mM of CaCl_2_, and 100 nM of dexamethasone (all Sigma) as described [47].

### 4.4. Stimulation with NETs

All experiments with isolated NETs were performed in MEM-α with 5% FCS which was heat-inactivated at 70 °C for 30 min to inactivate DNase (NET incubation medium [48]). The transfer of neutrophils was excluded by nuclear staining and immunofluorescence staining and the effect of a potential PMA contamination was analyzed.

#### 4.4.1. Resazurin Conversion

A 0.0025% Resazurin solution in plain MEM-α was added to the cells and fluorescence was measured at Ex 544/Em 590 nm with the Omega Plate reader (BMG Labtech, Ortenberg, Germany) after 60 min of incubation [49].

#### 4.4.2. SRB Staining

After the Resazurin measurement, cells were fixed with 99% ethanol at −20 °C for at least 1 h. After washing once with tap water, the wells were incubated with 0.4% SRB solution (Sigma-Aldrich, MO, USA) for 30 min. Cells were washed 3–4 times with 1% acetic acid. Staining was resolved with 10 mM of unbuffered Tris solution and absorbance was measured at 565 nm and 690 nm with the Omega plate reader. The absorbance values at 690 nm (impurities) were subtracted from the values at 565 nm [50].

#### 4.4.3. LDH Measurement

The release of lactate dehydrogenase (LDH) into the cell culture supernatant was measured by CyQUANT™ LDH Cytotoxicity Assay (Thermo Scientific, Waltham, MA, USA), according to the manufacturer’s instructions.

### 4.5. Cell Death Analysis

To investigate the cell death mechanism(s) induced by NETs, SCP-1 cells were incubated with Calcein-AM (2 μm; Biomol, Hamburg, Germany) for 50 min, to load them with fluorescent and cell-wall impermeable Calcein. Cells were washed twice to remove Calcein-AM. Then, NETs were added at a concentration of 0.5 ng/μL in NET incubation medium together with propidium iodide (1 μg/mL). Cells were incubated at 37 °C, 5% CO_2_ in a humidified atmosphere. Cells treated with 0.1% Triton-X-100 served as a positive control. Fluorescent images were taken with an EVOS FL microscope in the GFP (Ex 470 nm/Em 525 nm) and RFP (Ex 531 nm/Em 593 nm) channels (Thermo Scientific, Waltham, MA, USA). The supernatant was collected and the green fluorescence of leaked Calcein (Ex 470 nm/Em 525 nm) was measured with the ClarioStar plate reader (BMG Labtech).

### 4.6. Migration Assay

Migration assays were performed as previously described [50]. Briefly, SCP-1 cells were seeded at a density of 20,000 cells/well in a 96-well plate with silicon inserts. After cell attachment overnight, the silicon inserts were removed and the medium was changed to NET incubation medium containing different concentrations of NETs (0.03125–0.5 ng/μL). Microscopic images were taken immediately at 0 h. After 45 h, cells were stained with SRB for better visualization, and microscopic images were obtained (due to the risk of overgrowth in this assay, the incubation time was reduced from 48 h to 45 h), with the previously free area in the center. The percentage of remaining free area was calculated with the following formula: (1)$$\mathrm{Migration}\;[\%]=\frac{{\mathrm{Cell}-\mathrm{free}\;\mathrm{area}}_{45\;h}}{{\mathrm{Cell}-\mathrm{free}\;\mathrm{area}}_{0\;h}}\times 100\%$$

### 4.7. Conditioning of NETs

#### 4.7.1. DNase

For the destruction of DNA in the NETs, DNase (200 U/mL) was added together with the NETs onto the SCP-1 cells. Digestion of DNA was controlled by PCR. Supernatants of SCP-1 cells were collected after incubation and directly added to a PCR master mix (Red HS Master Mix (Biozym, Hessisch Oldendorf, Germany), Primer UGT1A6 forward: TGG TGC CTG AAG TTA ATT TGC T, reverse: GCT CTG GCA GTT GAT GAA GTA). PCR was done with the following conditions: 2 min initial denaturation, 35 cycles: 15 s 95 °C denaturation, 15 s 60 °C annealing, 15 s 72 °C elongation; 10 min 72 °C final elongation). Products were loaded on 1.8% agarose gel with 0.007% ethidium bromide (Carl Roth) and visualized with an INTAS GelDoc (INTAS, Göttingen, Germany). Band intensities were quantified with the ImageJ Gel Analysis tool.

#### 4.7.2. Protease Inhibitors

Proteinase K (0.5 μg/mL) or Protease inhibitors (Leupeptin 5 μg/mL, Pepstatin A 1 μg/mL) were added together with the NETs onto the SCP-1 cells, respectively.

#### 4.7.3. Heat-Conditioning

NETs were directly incubated at 40 °C for 1 h, at 75 °C for 20 min, or 99 °C for 10 min. Afterwards, NETs were diluted with NETs incubation medium.

#### 4.7.4. Heparin Treatment

Unfractionated heparin (LEO Pharma, Neu-Isenburg, Germany) was added together with NETs on SCP-1 cells in a concentration of 50 μg/mL [39,40].

#### 4.7.5. Histone Antibody Treatment

Histone antibodies were added together with NETs on SCP-1 cells (H2A CST #12349, H4 CST #2935). Both antibodies were used in a concentration of 50 ng/mL as previously published [38].

### 4.8. Statistical Analysis

Significance was calculated using a nonparametric test. If two groups were compared, the Mann–Whitney-U test was used, while for a comparison of several groups, the ANOVA Kruskal–Wallis test was applied and the Bonferroni correction was used for multiple comparisons. In addition, for the analysis of two different variables, the two-way ANOVA was used. All statistical analyses were performed with GraphPad Prism Version 8 (San Diego, CA, USA). A p-value below 0.05 was considered significant. All data are shown as box plots with median, interquartile range, and 95% confidence interval (Tukey modification to show outliers, indicated as black dots) if not indicated differently. The number of experiments is given in the figure legend (“N” independent experiments with NETs from different donors; “n” number of technical replicates).

## Figures and Tables

**Figure 1 ijms-23-16208-f001:**
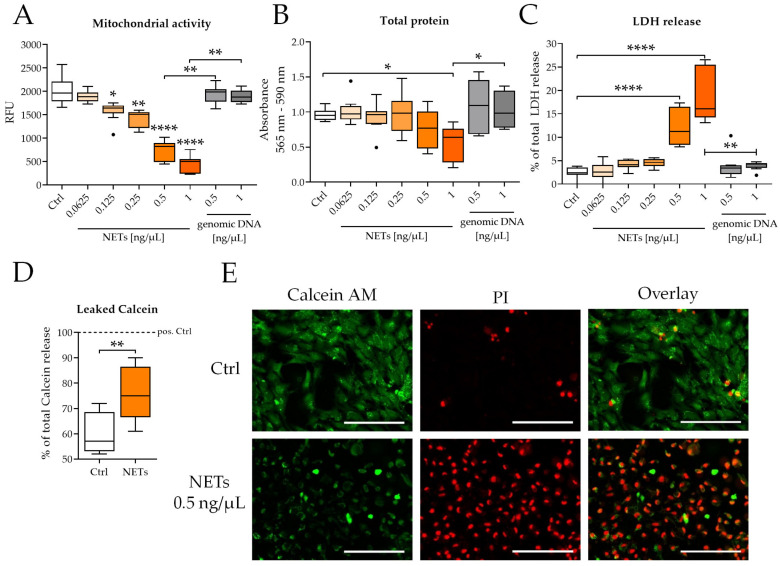
NETs are toxic to SCP-1 cells. SCP-1 cells were incubated with different concentrations of NETs for 48 h. (**A**) Mitochondrial activity was determined by Resazurin conversion. (**B**) Total protein content was determined by SRB staining. (**C**) Percental LDH release as a marker for cell death. (**D**) % of total Calcein released into the supernatant after NETs (0.5 ng/μL) treatment. Mann–Whitney U-test. (**E**) Live (Calcein-AM)/Dead (PI) staining of SCP-1 cells after 22 h. Scale bar 200 μm. N = 3, *n* = 3. * *p* < 0.05, ** *p* < 0.01, **** *p* < 0.0001 as determined by the Kruskal–Wallis test.

**Figure 2 ijms-23-16208-f002:**
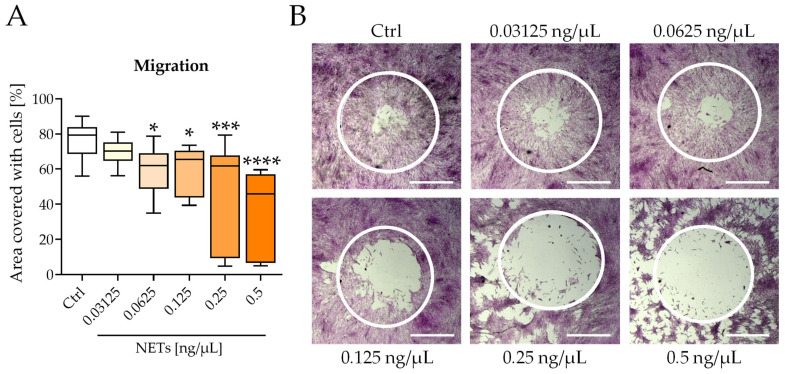
NETs impaired the migration of SCP-1 cells. The migration of SCP-1 cells was analyzed after 45 h of incubation with NETs. (**A**) Migration to a cell-free area. (**B**) Exemplary images of migration of SCP-1 cells after 45 h. Cells were stained with SRB. White circles highlight the cell-free area at time point 0 h. Original magnification 2×, scale bar 1000 μm. N = 3, *n* = 3. * *p* < 0.05, *** *p* < 0.001, **** *p* < 0.0001, as determined by the Kruskal–Wallis test.

**Figure 3 ijms-23-16208-f003:**
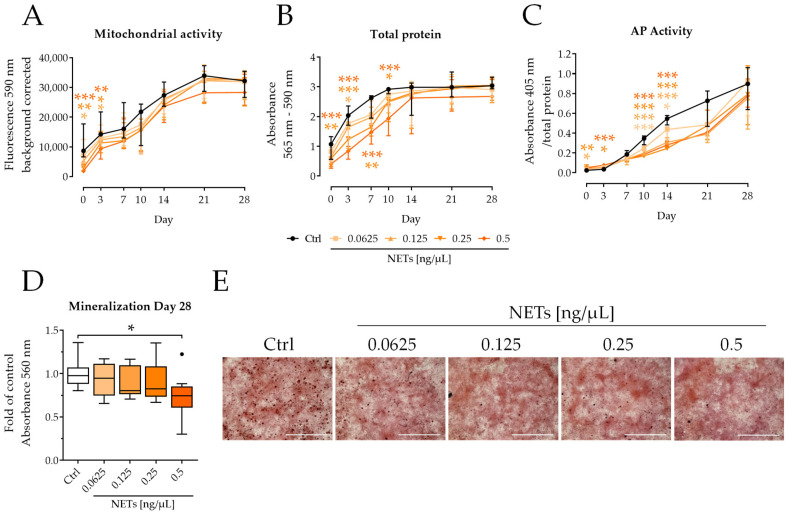
NETs impaired the differentiation of SCP-1 cells. SCP-1 cells were pre-incubated for 48 h with different concentrations of NETs, then osteogenically differentiated for 28 days. (**A**) Mitochondrial activity was determined by Resazurin conversion. (**B**) Total protein content was determined by SRB staining. (**C**) AP activity was determined photometrically. (**D**) Matrix mineralization was determined by Alizarin Red staining on day 28 of differentiation. (**E**) Exemplary images of Alizarin Red staining on day 28. Original magnification 2×, scale bar 2000 μm. N = 4, *n* = 3. Median ± 95% CI. * *p* < 0.05, ** *p* < 0.01, *** *p* < 0.001, as determined by two-way ANOVA.

**Figure 4 ijms-23-16208-f004:**
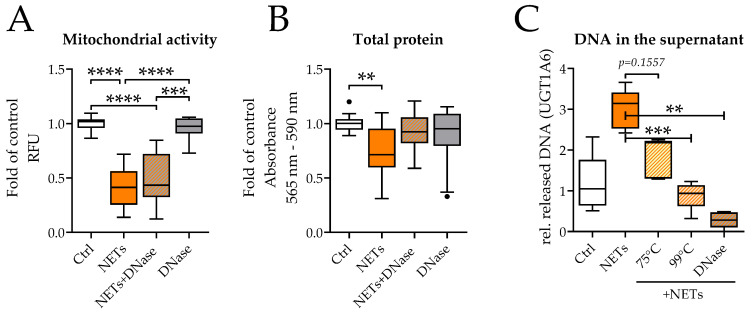
DNase did not prevent NETs toxicity. SCP-1 cells were incubated with NETs and DNase for 48 h. (**A**) Mitochondrial activity was determined by Resazurin conversion. (**B**) Total protein content was determined by SRB staining. (**C**) Effectiveness of DNA removal was analyzed by PCR. N = 3, *n* = 3. ** *p* < 0.01, *** *p* < 0.001, **** *p* < 0.0001, as determined by the Kruskal-Wallis test.

**Figure 5 ijms-23-16208-f005:**
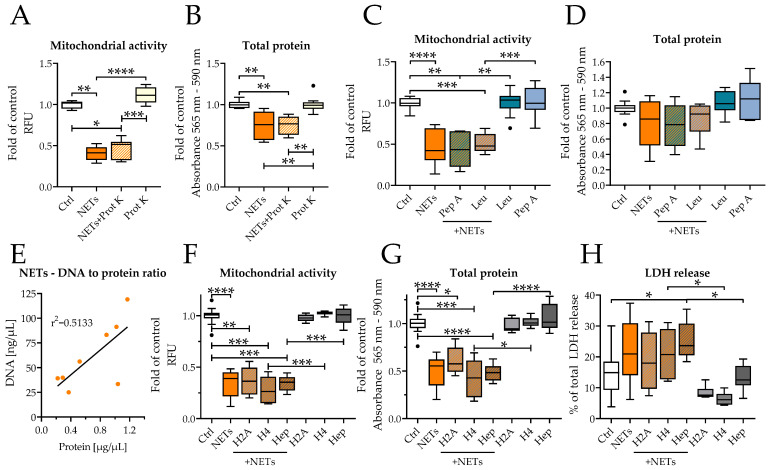
Inhibition of proteases or histones did not prevent NETs toxicity. (**A**,**B**) SCP-1 cells were incubated with 0.5 ng/μL of NETs and 0.5 μg/mL of Proteinase K. After 48 h, mitochondrial activity (**A**) and total protein content (**B**) were determined. (**C**,**D**) SCP-1 cells were incubated with 0.5 ng/μL of NETs and either 5 μg/mL of Leupeptin or 1 μg/mL of Pepstatin A. Mitochondrial activity (**C**) and total protein content (**D**) were determined. (**E**) The DNA content of NETs correlates with the protein content N = 3, *n* = 3. Effect of histone antibodies or heparin (50 μg/mL) treatment (**F**–**H**): (**F**) mitochondrial activity (resazurin conversion) values; (**G**) total protein content (SRB staining); and (**H**) LDH release as indicator for cell death (in relation to lysed cells). N = 2–3, *n* = 3. * *p* < 0.05, ** *p* < 0.01, *** *p* < 0.001, **** *p* < 0.0001, as determined by the Kruskal–Wallis test.

**Figure 6 ijms-23-16208-f006:**
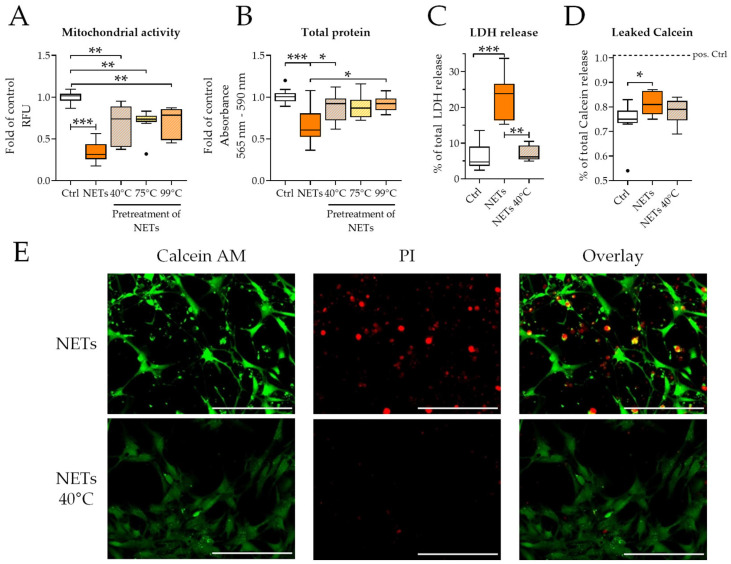
40 °C is sufficient to block NETs toxicity. NETs were incubated at the indicated temperatures before addition to SCP-1 cells for 48 h. (**A**) Mitochondrial activity determined by Resazurin conversion measurement. (**B**) Total protein content determined by SRB staining. (**C**) Percental LDH release as a marker for cell death. (**D**) Release of Calcein into the SCP-1 cell culture supernatant after 22 h of incubation, pos. Ctrl is Triton-X-100-treated cells. (**E**) Calcein-AM/PI staining of SCP-1 cells after 22 h. Scale bar 200 μm. N = 3, *n* = 3. * *p* < 0.05, ** *p* < 0.01, *** *p* < 0.001, as determined by the Kruskal–Wallis test.

## Data Availability

The datasets generated during and/or analyzed during the current study are available from the corresponding author upon reasonable request.

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
