# Peer review of "Febrile-Range Hyperthermia Can Prevent Toxic Effects of Neutrophil Extracellular Traps on Mesenchymal Stem Cells"

_ijms, 2022, doi:10.3390/ijms232416208_

Round 1

Reviewer 1 Report

in the file

Reviewer 2 Report

The present study shows that NETs are involved in detrimental effects in immortalized bone marrow derived mesenchymal stem cells (SCP-1), decreasing mitochondrial activity, increasing cell death and augmenting supernatant content of calcein. Also they described that NETs impaired SCP-1 migration and osteogenic differentiation. All these deleterious effects provoked by NETs were not reduced in the presence of DNAse or protease inhibitors, but interestingly, high  temperature (40°C -99°C) reverted the effects caused by NETs. With all these data, the authors propose that hyperthermia could have a beneficial effect on fracture healing inhibiting the negative effects of NETs in SCP-1 cells.

Major suggestions

The study is well designed and conducted. Although the authors demonstrated robustly the effects of NETs in SCP-1, it is necessary that they show the effect achieved using only PMA (500 nM) in SCP-1 cells. As described in methodology, isolation of NETs is from neutrophils stimulated with PMA 500 nM for 4 h. Thus, NETs used in the present research can contain PMA, then this assay is mandatory to describe (if any) the possible effects of PMA in SCP-1 cells.

Minor suggestions

The authors claim the following:

“As seen before, mitochondrial activity was significantly reduced at the three highest (117)

concentrations of NETs (0.125-0.5 ng/µL). This effect persisted until day 3. After day 7 (118)

mitochondrial activity has nearly recovered in all conditions (Figure 3A), suggesting a (119)

prolonged negative effect of NETs on SCP-1 cells.” (120)

I personally disagree because this is a transitory effect instead of a “prolonged negative effect”.

Figure 4 should be carefully revised. The authors mentioned treatments such as proteinase K treatment which is not mentioned in the main text and in my opinion figures 4 and 5 are in some point confusing. 

Line 209: “In general, prevention of NET formation would be the more favorable approach but…” change “ …more …” to “… most …”

Round 2

Reviewer 2 Report

The authors answered clearly and sufficiently all the queries solicited. I reccommend its publication in the current form.